# *SERPINA1* Peptides in Urine as A Potential Marker of Preeclampsia Severity

**DOI:** 10.3390/ijms21030914

**Published:** 2020-01-30

**Authors:** Natalia Starodubtseva, Natalia Nizyaeva, Oleg Baev, Anna Bugrova, Masara Gapaeva, Kamilla Muminova, Alexey Kononikhin, Vladimir Frankevich, Eugene Nikolaev, Gennady Sukhikh

**Affiliations:** 1National Medical Research Center for Obstetrics, Gynecology and Perinatology Named after Academician V.I. Kulakov of the Ministry of Healthcare of Russian Federation, 117997 Moscow, Russia; n_starodubtseva@oparina4.ru (N.S.); niziaeva@gmail.com (N.N.); o_baev@oparina4.ru (O.B.); a_bugrova@oparina4.ru (A.B.); m_gapaeva@oparina4.ru (M.G.); k_muminova@oparina4.ru (K.M.); konoleha@yandex.ru (A.K.); bainat26@yandex.ru (G.S.); 2Moscow Institute of Physics and Technology, 141701 Moscow, Russia; 3Emanuel Institute for Biochemical Physics, Russian Academy of Sciences, 119991 Moscow, Russia; 4Skolkovo Institute of Science and Technology, Skolkovo, 121205 Moscow, Russia; 5V.L. Talrose Institute for Energy Problems of Chemical Physics, Russian Academy of Sciences, 119991 Moscow, Russia; ennikolaev@gmail.com; 6First Moscow State Medical University Named after I.M. Sechenov, 119146 Moscow, Russia

**Keywords:** preeclampsia, diagnostics, proteomics, mass spectrometry, alpha-1 antitrypsin, syncytiotrophoblast

## Abstract

Preeclampsia (PE) is a multisystem disorder associated with pregnancy and its frequency varies from 5 to 20 percent of pregnancies. Although a number of preeclampsia studies have been carried out, there is no consensus about disease etiology and pathogenesis so far. Peptides of *SERPINA1* (α1-antitrypsin) in urine remain one of the most promising peptide markers of PE. In this study the diagnostic potential of urinary α1-antitrypsin peptides in PE was evaluated. The urinary peptidome composition of 79 pregnant women with preeclampsia (PE), chronic arterial hypertension (CAH), and a control group was investigated. Mann–Whitney U-test (*p* < 0.05) revealed seven PE specific *SERPINA1* peptides demonstrating 52% sensitivity and 100% specificity. *SERPINA1* in urine has been associated with the most severe forms of preeclampsia (*p* = 0.014), in terms of systolic hypertension (*p* = 0.01) and proteinuria (*p* = 0.006). According to Spearman correlation analysis, the normalized intensity of *SERPINA1* urinary peptides has a similar diagnostic pattern with known diagnostic PE markers, such as sFLT/PLGF. *SERPINA1* peptides were not urinary excreted in superimposed PE (PE with CAH), which is a milder form of PE. An increase in expression of *SERPINA1* in the structural elements of the placenta during preeclampsia reflects a protective mechanism against hypoxia. Increased synthesis of *SERPINA1* in the trophoblast leads to protein accumulation in fibrinoid deposits. It may block syncytial knots and placenta villi, decreasing trophoblast invasion. Excretion of PE specific *SERPINA1* peptides is associated with syncytiotrophoblast membrane destruction degradation and increased *SERPINA1* staining. It confirms that the placenta could be the origin of *SERPINA1* peptides in urine. Significant correlation (*p* < 0.05) of *SERPINA1* expression in syncytiotrophoblast membrane and cytoplasm with the main clinical parameters of severe PE proves the role of *SERPINA1* in PE pathogenesis. Estimation of *SERPINA1* peptides in urine can be used as a diagnostic test of the severity of the condition to determine further treatment, particularly the need for urgent surgical delivery.

## 1. Introduction

Preeclampsia (PE) is a multisystem disorder associated with pregnancy and its incidence varies from 5% to 20% of pregnancies. PE is a major cause of maternal mortality, preterm birth, and morbidity [1]. PE manifests with high blood pressure (systolic ≥140 mmHg or diastolic ≥90 mmHg) and proteinuria ≥0.3 g/L accompanied with kidneys and liver dysfunction and fetal growth restriction, often with edema [1]. PE increases the risk of adverse pregnancy outcomes significantly, such as cesarean section, difficult labor, and preterm child birth [2]. Early delivery is the most efficient treatment for severe PE [3]. The doctor has to make a difficult decision: prolong the pregnancy or deliver if PE is suspected. Prolongation significantly increases the child’s chances of survival (especially with an early form of pathology—up to 34 weeks) but can cause irreparable harm to the body of a woman. Women who have undergone preeclampsia are predisposed to the development of arterial hypertension, heart disease, and stroke [4,5,6,7]. The prediction of PE still remains extremely difficult [8], making the search for accurate markers of preeclampsia one of the key tasks of personalized medicine, in particular to accurately determine the timing of delivery.

The pathophysiology of this disorder is still unclear. The most studied cause of PE is the incomplete transformation of the spiral arteries due to weak trophoblast invasion [9]. 

Early studies of PE diagnostics were dedicated to the discovery of biomarkers in various biological media, such as serum. For example, the pregnancy-specific beta-1-glycoprotein 2 (*PSG2*) protein was identified using isobaric tags for relative and absolute quantitation [2]. *PSG2* plays a key role in lipid metabolism and inflammation. Urine is another promising biomaterial for such studies. Three potential urinary PE biomarkers were found including serotransferrin (*TF*), complement factor B (*CFB*), and serum paraoxonase/arylesterase 1 (*PON1*) by 2-d LC-MS/MS with ELISA validation [3]. Another promising approach is based on a peptidome analysis of the biological fluids of women with PE [10,11,12,13,14]. Peptide analysis of serum samples revealed to panel of peptides, including peptides of fibrinogen alpha (*FGA*), *α*1-antitrypsin (*A1AT*), apolipoprotein L1 (*APO-L1*), inter-alpha-trypsin inhibitor heavy chain H4 (*ITIH4*), kininogen-1 (*KNG1*), and thymosin beta-4 (*TMSB4*), that can be used for PE diagnosis [12]. Peptidome fingerprinting revealed unique urine peptides for preeclampsia differentiating from other hypertensive or proteinuric disorders (chronic arterial hypertension, acute/chronic pyelonephritis) in pregnancy [11,12,13]. Previous studies showed that 35 peptides reliably distinguished a particular PE group (severe or mild) from control group [13]. The results revealed unique peptides of *α*1-antitrypsin, collagen alpha-1(I) chain, collagen alpha-1 (III) chain, and uromodulin that can potentially serve as early indicators of PE. 

The ability of these peptides in preeclamptic urine to bind the amyloidophilc dye Congo Red was successfully applied for clinical practice [15]. In the presence of Congo red azo-dye, the amyloid-like structures aggregate in urine [16]. It was shown that peptides of b-sheets, *α*1-antitrypsin (*SERPINA1*), complement 3, haptoglobin, ceruloplasmin, and trypstatin are most likely targets for Congo red binding [16]. A no-invasive Congo Red Dot Paper Test for rapid identification of PE was also presented [17]. *α*1-antitrypsin is one of the most promising peptide markers of PE that have significantly higher aggregation capacity [16,18,19]. *SERPINA1* is an abundant plasma protein primarily expressed in liver and involved in serine proteases inhibition, primarily neutrophil elastase. It also inhibits trypsin, chymotrypsin, and plasminogen activator [20]. According to the previous studies, *SERPINA1* peptides have higher serum, urine, and placental immunoreactivity [11]. The result also shows that the increase of *SERPINA1* also can be associated with acute inflammatory conditions. The hypothesis suggests that *SERPINA1* fragments have the propensity to misfold and aggregate into supramolecular structures during PE [21]. This allows attributing preeclampsia to a wide cohort of pathologies associated with pathological aggregation of a particular protein (in particular, *SERPINA1*). The list of protein conformational disorders includes Alzheimer’s disease, fetal growth restriction, diabetes, chronic autoimmune, cardiovascular or renal diseases, and aging [16,21,22,23,24].

The aim of this research was to evaluate the diagnostic potential of urinary *SERPINA1* peptides for PE and to study the features of *SERPINA1* expression in the placenta in hypertensive disorders of pregnancy.

## 2. Results

### 2.1. Clinical Data

The study included 79 pregnant women who were divided into three groups: the first group included 18 pregnant women with an uncomplicated pregnancy; the second group included 17 pregnant women with chronic hypertension (CAH); and the third group included 44 pregnant women with preeclampsia (PE). Eight women of the PE group developed this pathology under initial chronic arterial hypertension (PE with CAH), and there were no previous episodes of arterial hypertension in the remaining 36 pregnant women. Severe PE was diagnosed for 23 women; 10 women had an early form of the disease.

Patients included in the study did not differ in age. However, body mass index was higher in women with CAH and PE. Primiparity in the group of uncomplicated pregnancy amounted to 55.5%, in the group of CAH to 47.1%, and in the group of PE to 70.4%. Analysis of the outcome of previous pregnancies and childbirth showed that preeclampsia in a previous pregnancy was more common in women with CAH (23.5%) and PE (18.2%) (Table 1).

Comparison of clinical and laboratory examination data at the time of diagnosis showed that the level of proteinuria (PU) in the PE group was significantly higher than in women with uncomplicated pregnancy and CAH. The highest numbers of blood pressure occurred in the group of women with PE. The frequency of delivery by caesarean section did not differ in the groups. In all observations, live children were born with an Apgar score of 1 to 9.

In the PE group, a low Apgar score at the time of birth was significantly more likely to occur, and newborns often needed treatment in the intensive care unit (Table 1).

### 2.2. SERPINA1 Peptides in Urine

The peptide composition of urine samples collected from 79 patients (control—18, CAH—17, PE—44) was analyzed. Due to the large number of possible PTMs (methionine, proline, and lysine oxidation) as well as nonspecific cleavage on both the -N and -C terminal groups database, peptide searching required significant computational resources. To facilitate the search procedure, a small database for 145 proteins was previously created [13,16]. The group of PE patients was formed without any stratification by the period of clinical manifestation and health condition severity due to the small number of patients in the subgroups.

A total of 1542 peptides were detected as a result. The peak intensities of each peptide were normalized to the sum of the intensities of the peptides in each specific sample. A total of 248 peptides present in each group minimum in five different samples with a MS/MS count of ≥3 (Appendix A): on average 111 ± 25 peptides in each sample. *SERPINA1* peptides (*n* = 50) take the second place in the peptide occurrences after the collagen-alpha-1 (I) chain among them. These peptides belong to the most functionally interesting part of the urine peptidome associated with hypertensive disorders. Pairwise Mann–Whitney U-test (*p* < 0.05) revealed seven *SERPINA1* peptides specific for PE, which were not detected in any of the urine samples of CAH and control groups (Figure 1).

All but one *SERPINA1* peptide are located in the C terminal and only one peptide (LRTLNQPDSQLQLTTGNGLF) in the central part (Figure 1). These peptides were associated with severe PE previously [11,13,16,21]. C-terminal *SERPINA1* fragment has aggregation ability significantly higher than the 16-KLVFF-20 fragment of beta-amyloid, supporting the theory of PE as a protein conformational disorder.

However, PE specific peptides were detected only in 23 urine PE samples. The results demonstrate high specificity (100%), but moderate sensitivity (52%) of peptide biomarkers. The PE patients were divided into two groups: with *SERPINA1* peptides in urine (PE_SER+) and without (PE_SER−). 

### 2.3. Clinical Features of PE Cases with Urinary SERPINA1 Peptides

*SERPINA1* peptides were not found in the urine of women with uncomplicated pregnancy, chronic arterial hypertension, or preeclampsia on the background of CAH.

Clinical differences between the PE groups with/without *SERPINA1* peptides in urine (PE_SER+/PE_SER−) are associated with the level of systolic blood pressure, proteinuria, severe proteinuria, and the frequency of severe forms of preeclampsia (Table 2). Moreover, the most pronounced violations occurred in the PE_SER+ group. Urinary excretion of *SERPINA1* peptides was not associated with the duration of preeclampsia and the incidence of fetal growth retardation.

A correlation analysis of clinical data is shown in Figure 2. The group parameter corresponds to the severity of a woman’s condition: 0—control, 1—CAH, 2—PE-CAH, 3—mild PE, 4—severe PE. This parameter correlates with almost all significant clinical and laboratory indicators of hypertensive pregnancy disorders (Figure 2). The level of urinary *SERPINA1* peptides positively correlated with the disease severity (parameter–group) with *r*_s_ = 0.48. The opposite tendency was found for *SERPINA1* expression in placenta (*r*_s_ = −0.56). This confirms our hypothesis concerning the diagnostic potential of urinary *SERPINA1* peptides. The level of sFLT/PLGF (the only known marker of PE) correlated significantly (*p* < 0.05) with systolic (*r*_s_ = 0.81)/diastolic blood pressure (*r*_s_ = 0.63), proteinuria (r_s_ = 0.76), the markers of liver damage (positively with ALT (*r*_s_ = 0.40), AST (*r*_s_ = 0.45), lactate dehydrogenase (LDH, *r*_s_ = 0.59)), kidney damage (positively with urea (*r*_s_ = 0.63) and negatively with glomerular filtration rate (GFR, *r*_s_ = −0.5)), hemostasiogram violation (negatively with prothrombin time (r_s_ = −0.66) and international normalized ratio (r_s_ = −0.56), positively with prothrombin index (r_s_ = 0.56)), and impaired feto-placental blood flow (positively with uterine artery (r_s_ = 0.57) and umbilical artery (r_s_ = 0.63) pulsatility indexes). sFLT/PLGF negatively correlated with the babys’ health after delivery (Apgar score (r_s_ = −0.43)), the newborns’ weight (r_s_ = −0.57), and height (r_s_ = −0.56), newborns’ time under intensive care (ICU (days), (r_s_ = −0.59)). Average intensity of *SERPINA1* peptides excreted in urine correlated (*p* < 0.05) positively with blood pressure (r_s_ = 0.30), proteinuria (r_s_ = 0.30), ALT (r_s_ = 0.33), AST (r_s_ = 0.37), LDH (r_s_ = 0.36), sFLT/PLGF (r_s_ = 0.33), newborns’ time under intensive care (r_s_ = 0.37), and negatively with GFR (r_s_ = −0.40). Thus, *SERPINA1* peptides intensity in urine has a similar diagnostic pattern with known diagnostic PE markers such as sFLT/PLGF.

### 2.4. SERPINA1 Expression in Placenta

The histological examination of placenta sections stained with haematoxylin and eosin from the uncomplicated full-term pregnancies showed normally capillarized villous trees with a balance of mature intermediate and terminal villi. Immature intermediate villi were present as small clusters. All stem villi were completely formed. Histological examination of placenta sections from the CAH group showed that the villi met the gestational age. The type of angiogenesis with the presence of hypercapillarized terminal villi prevailed. Dystrophic changes in villi were less pronounced. There is a moderate amount of fibrinoid deposits. The number of syncytial knots was increased. 

In patients suffering from PE groups there were perivillous fibrinoid deposits, multiple infarcts of the villus tree, with formation of afunctional zones excluded from the blood circulation (Figure 3). In these areas, an increased number of syncytial knots were noted, being a manifestation of compensatory mechanisms. Syncytial knots are considered to foci of trophoblast proliferation in the form of hyperchromic nuclei surrounded by a common cytoplasm located on the surface of the placental villi. The number of syncytial knots was increased. The amount of fibrinoid deposits was larger in cases of preeclampsia than in an uncomplicated pregnancy.

In placental sections in PE patients with *SERPINA1* peptides in urine (PE_SER+ group), we found in intermediate villi (mature and immature) with evident dystrophic changes, trophoblast shedding from the villi surface, and focal stromal fibrosis. Capillaries in intermediate villi were irregularly located and avascular villi were also present. Multiple villous tree infarctions were detected. In the PE_SER− group, dystrophic changes were less noticeable, and a branched type of angiogenesis predominated as well as in the CAH group. Stromal canals formed by mesenchyme cells were preserved.

Immunohistochemical study (staining with primary antibodies to *SERPINA1*) in control group showed weak staining in the cyto- and syncytiotrophoblast of different types of placental villi (stem, mature, terminal villi), invasive trophoblast, syncytial knots decidual cells, single mesenchymal stromal cells, placental macrophages (Kashchenko–Hoffbauer cells), and amniotic epithelium. No reaction was observed in perivillous fibrinoid (Figure 4C–H). *SERPINA1* expression was significantly increased in PE cyto- and syncytiotrophoblast, syncytial knots, extravillous trophoblast, and decidual cells. In this case, the most pronounced staining was present in fibrinoid deposits (Rohr and Nitabuh fibrinoid), in the region of the decidual plate and perivillous (Figure 3D,F,H and Figure 4B,F,G,H). 

Figure 5 shows the differences in the expression of *SERPINA1* in the placental villous tree of PE_SER+ and PE_SER− groups. It should be noted that membrane staining prevails. In contrast, cytoplasmic staining of cyto- and SCT dominated in the PE_SER+ group. However, in a number of observations, *SERPINA1* staining was absent due to pronounced dystrophic changes and desquamation of the trophoblast villi (Figure 3A–H).

*SERPINA1* stained granules were observed in cyto- and SCT in all groups (Figure 3A–H and Figure 4A–H). The number of granules had a tendency to increase in cases of PE_SER- and CAH groups (Figure 4A,B). The granule size was significantly higher in PE_SER− and CAH (*p* < 0.001) groups compared to PE_SER+ (Figure 5A).

Significant correlation (*p* < 0.05) was found for *SERPINA1* expression in SCT membrane and cytoplasm with systolic (r_s_ = −0.54) and diastolic (r_s_ = −0.45) pressure, proteinuria level (r_s_ = −0.56), urea (r_s_ = −0.35), GFR (r_s_ = 0.45), PLGF (r_s_ = 0.63), sFLT (r_s_ = −0.44), sFLT/PLGF (r_s_ = −0.50), prothrombin time (r_s_ = 0.46), uterine (r_s_ = −0.37) and umbilical artery (r_s_ = −0.41) pulsatility indexes, child weight (r_s_ = 0.45) and height (r_s_ = 0.42), and newborns’ stay in intensive care (r_s_ = −0.45) (Figure 2). 

## 3. Discussion

The weakest trophoblast invasion and incomplete transformation of the uterine spiral arteries are the most studied causes of preeclampsia. It is known that the level of trophoblast invasion can be different (from the maximum in cases of ingrowth of the placenta villi into the uterine wall to the minimum presented during preeclampsia). One of these mechanisms governing trophoblast invasion may be an expression of *SERPINA1*. It is known that with pre-eclampsia, the level of proteolytic enzymes in particular metalloproteinases (*ADAM 12*) in the serum samples and placental tissue decreases [25,26]. One of the reasons for this may be a significant increase in the protease inhibitor. The presence of *SERPINA1* is a physiological mechanism that regulates trophoblast invasion and the transformation of spiral arteries, preventing excessive invasion of trophoblast villi into the uterine wall in our opinion. In the case of PE, the level of *SERPINA1* rises, especially in the cyto- and syncytiotrophoblast, as well as in the decidual plate. At the same time, the amount of fibrinoid deposits increases, resulting in the reduction of intervillous space, structural “blocking” of trophoblast, limiting trophoblast invasion into the uterine spiral arteries. It further promotes the disturbances of feto-placental blood flow, leading to a vicious circle. α1-antitrypsin inhibits one of the most common proteases—trypsin. It indicates changes in the metabolic pathways of protein degradation occurring during PE. In addition, family members of *SERPINA1* prevent collagen breakdown, which triggers hypoxia [27]. In the culture of trophoblast cells under hypoxia, there is an increase in the expression of the *SERPINA1* gene by three times [28]. Both HELLP syndrome and preeclampsia are characterized by a systemic pro-inflammatory response [29]. *SERPINs* inversely correlates with the production of pro-inflammatory cytokines and inhibits the development of a pro-inflammatory response [30]. 

An increase in the concentration of *SERPINs* in blood plasma was detected in the HELLP syndrome in which thrombocytopenia and disorders associated with the blood coagulation system play a leading role [31]. Fibrinoid deposits are formed at the foci of placental damage, more often with coagulation of blood plasma proteins. It is known that fibrinoid may be of two types. One of them is formed from blood components and after platelet destruction (fibrin type) [32], especially in the case of Rohr and Nitabuh fibrinoid. The other is a product of the trophoblast synthesis (matrix type), containing laminins, sialic acids, collagens, and heparan sulfate [33]. In most cases, fibrinoid consists of different types. The protective function of the fibrinoid is associated with the masking effect of antigens, preventing the immune conflict between the mother and child. The damage of trophoblast leads to the unmasking of antigenic determinants and the activation of inflammatory signaling cascades. In addition, fibrinoid, like a sponge, can adsorb a number of antigens, lowering the immune load [33]. However, a significant increase of fibrinoid itself becomes a problem leading to gluing of the villi, reducing of the intervillous space, a decrease in blood circulation, and promoting the placental insufficiency. Thus, the placental structures can be a source of *SERPINA1*. The possibility of an accumulation of *SERPINA1* in the fibrinoid was also noted. Under normal conditions, Rohr and Nitabuh fibrinoids locate around an invasive trophoblast, changing the degree of invasion and its activity. The accumulation in *SERPINA1* in fibrinoid deposits may be a new mechanism for the regulation of the trophoblast invasion degree [34]. 

Electron microscopy showed that SCT membrane staining of the placenta villi decreased significantly with the increase of pregnancy-related hypertension severity (in the following line: Control–CAH–PE) with a main change in PE_SER+. It can be explained by an impairment of the membranes and a reduction of microvilli on the syncytiotrophoblast surface, vacuolization of the cytoplasm, shedding to membranes and apical part of cells in severe PE cases [35].

Both *SERPINA1* membrane staining and cytoplasm *SERPINA1* stained granules in syncytiotrophoblast significantly decreased in the PE_SER+ group (Figure 5). Thus, placenta (mainly, trophoblast) could be the source of *SERPINA1* peptides in urine. Evaluation of the urinary *SERPINA1* peptides level may result from the destroyed trophoblast, a pathognomonic sign of severe preeclampsia, and the release of granules with *SERPINA1*. The maximum granule size was observed in SCT in the CAH group. 

Significant correlation (*p* < 0.05) of *SERPINA1* expression in syncytiotrophoblast membrane and cytoplasm with the main clinical parameters of severe PE (systolic/diastolic pressure, proteinuria level, GFR, PLGF, sFLT, sFLT/PLGF, prothrombin time, uterine and umbilical artery pulsatility indexes, newborns’ weight/height, and the duration of stay in newborns’ intensive care unit) proves the role of *SERPINA1* in PE pathogenesis. The decrease in the glomerular filtration rate in severe preeclampsia correlated with the level of *SERPINA1* peptides in the urine in PE along with sFLT/PLGF. The decrease in the glomerular filtration rate is one of the clinical manifestations of severe PE, along with protenuria. It can be assumed that by the presence of *SERPINA1* in the urine, two different forms of the clinical course of preeclampsia can be distinguished, differing in main clinical parameters (blood pressure, proteinuria, ALT, AST, LDH, GFR). PE specific *SERPINA1* peptides were not found in urine of patients with superimposed PE (PE associated with CAH). According to some authors, it was believed that PE against the background of CAH has a more favorable course [6,7]. Our study clearly showed that by the level of *SERPINA1* expression in urine, preeclampsia can be divided into two pathogenic variants: more favorable (with no *SERPINA1*) and less favorable with higher blood pressure, proteinuria, and a higher percentage of cardiopathology, and disorders in the system of the blood coagulation system (with *SERPINA1*) (Figure 2). Estimation of *SERPINA1* in urine can be used as a diagnostic test for the severity of preeclampsia with very high specificity. Moreover, urinary *SERPINA1* peptides detection is important to differentiate CAH and superimposed PE (PE with CAH) with real preeclampsia. This will clarify the management tactics of patients and the need for urgent surgical delivery.

## 4. Materials and Methods 

### 4.1. Experimental Design and Collection of Samples

Urine samples were obtained from 79 patients divided into three groups: preeclampsia (PE), 44 patients; chronic arterial hypertension (CAH), 17; and healthy pregnant women (control group), 18. Samples were collected from November 2018 to June 2019. All patients read and signed an informed consent approved by the Ethical Committee (Record No12 from 17 November 2016) of the National Medical Research Center for Obstetrics, Gynecology, and Perinatology, named after Academician V.I. Kulakov of the Ministry of Healthcare of Russian Federation. 

The inclusion criteria for this study were chronic arterial hypertension (group CAH) or systolic blood pressure ≥140 mm Hg or diastolic blood pressure ≥90 mm Hg on two occasions at least 4 h apart after 20 weeks of gestation in a woman with a previously normal blood pressure and proteinuria ≥0.3 g per 24 h urine collection (group PE) (according to ACOG Practice Bulletin number 202, 2019). Severe preeclampsia features included severe hypertension (systolic blood pressure ≥160 mm Hg or diastolic blood pressure ≥110 mm Hg); thrombocytopenia (platelet count less than 100 × 10^9^/L), elevated blood concentrations of liver enzymes (twice the upper limit of normal concentration), oliguria (500 mL of urine per day or less); epigastric pain or pain in right hypochondrium; pulmonary edema. Control group consisted of healthy pregnant women with uncomplicated course of gestation.

Patients with a manifestation of pathology up to 33 weeks were assigned to the early form of PE. The control group consisted of women with a physiological pregnancy and with a total protein concentration in the urine below 100 μg/mL. The exclusion criteria for this study were multiple pregnancy, pregnancy after assisted reproductive technology (ART), diabetes, transplanted organs, autoimmune diseases, oncological diseases, decompensated kidney disease, chromosomal abnormalities in the fetus, congenital malformations of the fetus, antenatal fetal death.

Urine samples were taken at the time of admission. All patients received appropriate treatment with antihypertensive drugs, magnesium sulfate, and glucocorticoids according to indications. However, the treatment was carried out after collecting urine samples. For peptidomic analysis, urine was collected after preliminary hygiene procedures. Urine was centrifuged for 10 min (2000 *g*, at 4 °C) to remove cell debris. The supernatant was stored at −80 °C until use.

### 4.2. LC-MS/MS Analysis 

Peptides were isolated by gel filtration and analyzed by high-performance liquid chromatography with tandem mass-spectrometry (HPLC-MS/MS) in full accordance with the protocols developed by the authors earlier [13,14,16]. A total of 1.5 mL of urine was mixed with 1.5 mL of denaturing buffer (4M urea, 20 mM NH4OH, 0.2% sodium dodecyl sulfate), the stages of ultrafiltration and gel filtration were carried out in order to separate proteins, purify low molecular weight contaminants, and change the buffer. Urine samples were analyzed in triplicate on a nano-HPLC Agilent 1100 system (Agilent Technologies, Santa Clara, CA, USA) combined with a 7-T LTQ-FT Ultra mass spectrometer (Thermo Electron, Bremen, Germany). Peptides separation was carried out by a 95-min gradient (solvent A—LC grade Water (H_2_O) containing 0.1% of formic acid/solvent B—LC grade acetonitrile (ACN) containing 0.1% of formic acid) from 3% to 35% of solvent B at a flow rate of 300 nl/min. ESI ion source was used in positive ion mode (2.3 kV). MS and MS/MS data were obtained in data-dependent mode using the Xcalibur (Thermo Finnigan, San Jose, CA, USA) software. The precursor ion scan MS spectra (*m/z* range 300–1600 a.m.u.) were acquired in the FTICR with resolution R = 50,000 at *m/z* 400 a.m.u. (number of accumulated ions: 5 × 10^6^). Obtained MS data analysis was processed using the PEAKS Studio software [36].

### 4.3. Histology and Immunohistochemistry of Placenta

After macroscopic examination, tissue fragments from the central zone of each placenta were dissected through the entire depth of placental disk and fixed in a 10% neutral formalin solution (Biovitrum, Russia; pH 7.4). Paraffin-embedded placental tissue sections (4 µm thick) were used for histological (stained with haematoxylin and eosin) [24] and immunohistochemical examination with primary polyclonal antibodies to *SERPINA1*.

Immunohistochemistry was performed on 4 μm thick paraffin sections using a Ventana Medical System closed type immunity tester (Roche, UK) with closed detection kit. For the visualization we used the Ultra View Universal DAB Detection Kit. The automated staining protocol included all steps of a standard immunohistochemistry procedure: sections wax removal, antigen unmasking, blocking of endogenous peroxidase, and incubation with primary and secondary antibodies. The study was carried out at +37 °C. Polyclonal antibodies to *SERPINA1* (1:500; PA5-26439, Invitrogen, Waltham, MA, USA) were used as primary antibodies. The brown staining of cytoplasm was considered as a positive immunohistochemical reaction. The intensity of *SERPINA1* immunohistochemical staining of syncytiotrophoblast membrane and cytoplasm was evaluated using an image analysis system in units of optical density based on a Nikon Eclipse microscope with the NIS-Elements software (Melville, NY, USA). In addition, the diameters of *SERPINA1* stained granules in the SCT cytoplasm (μm), as well as the number of granules in the SCT, were measured using the NIS-Elements program. The field of the view within the calculation was equal to 175 × 130 µm. 

As a negative control, the samples from the studied sections underwent a standard immunohistochemical analysis without incubation with primary antibodies. 

### 4.4. Urine Peptide Identification

Search and identification of peptides was carried out using the MaxQuant program (version 1.5.3.30, Munich, Germany) and using an artificially generated base of 145 proteins identified earlier [13,16]. The creation of the database is dictated by the need to reduce the time and resources. The following search parameters were used to identify peptides: nonspecific cleavage, variable modifications—oxidation of methionine, lysine and proline, up to 5 variable modifications per peptide were allowed, mass accuracy for the precursor ion—20 ppm, mass accuracy for MS/MS fragments—0.50, the minimum length of the peptide was 5 amino acids, the maximum mass of the peptide was 10 kDa, false detection rate (FDR) ≤0.01, and the minimum score for unmodified peptides was 20 and for the modified ones 40. Semi-quantitative results for each protein identified in the sample were obtained by the label-free method with alignment of the chromatograms by the time of peptide exit and normalization to the total intensity. The peak intensities of each peptide were normalized to the sum of the intensities of the peptides in each specific sample.

### 4.5. Statistical Analysis

Scripts written in the R language [37] and the RStudio program [38] were used for statistical processing of the results. Statistical analysis was performed using the Mann–Whitney test for pairwise comparison of groups with Bonferroni correction for multiple comparisons. Mean value, standard deviation, minimum and maximum values, median and interquartile interval, and frequencies (%) for qualitative data were used to describe the quantitative parameters. Differences between groups for qualitative data were calculated using non-parametric statistics, X^2^. The value of the threshold significance level *p*-value was taken equal to 0.05. For the convenience the level of *SERPINA1* expression in the placenta was multiplied by 100.

## 5. Conclusions

With an increase in expression of *SERPINA1*, the structural elements of the placenta during preeclampsia reflect a protective mechanism with an increase in protease inhibitors under hypoxia. Increased synthesis of *SERPINA1* in the trophoblast leads to protein accumulation in fibrinoid deposits, blocking syncytial knots and placenta villi, contributing to trophoblast invasion decrease and probably causing inferiority of the spiral arteries, as well as impaired development of the placental villus tree. 

In severe preeclampsia, as previously shown by electron microscopy, damage to the structures of the placenta, especially the trophoblast membranes, results in the release of *SERPINA1* granules from the cytoplasm of the cyto- and syncytiotrophoblast into the maternal duct. The destruction of cyto- and syncytiotrophoblast may explain the decrease in *SERPINA1* staining up to its complete disappearance. Trophoblast shedding leads to the penetration of granules containing *SERPINA1* into maternal bloodstream and may result in *SERPINA1* peptides appearance and/or increase in the urine. Urinary excretion of PE specific *SERPINA1* peptides was linked with the most severe forms of preeclampsia in clinical manifestations, primarily the level of systolic hypertension and proteinuria. These *SERPINA1* peptides were not detected in uncomplicated pregnancy and chronic arterial hypertension, preeclampsia with CAH. Thus, detection of *SERPINA1* in urine can be used as a diagnostic test for the severity of preeclampsia. Distinguishing between PE, PE with CAH, and CAH is particularly important for the management of these patients.

## Figures and Tables

**Figure 1 ijms-21-00914-f001:**
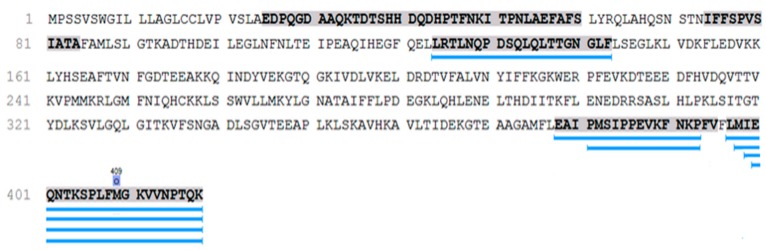
Amino acid sequence of the alpha-1-antitrypsin protein (*SERPINA1*). Underlining shows seven urinary *SERPINA1* peptides specific for preeclampsia according to pairwise Mann–Whitney U-test with Bonferroni correction (*p* < 0.05). In bold, PE specific peptides selected by I. Buchimshi are shown [11].

**Figure 2 ijms-21-00914-f002:**
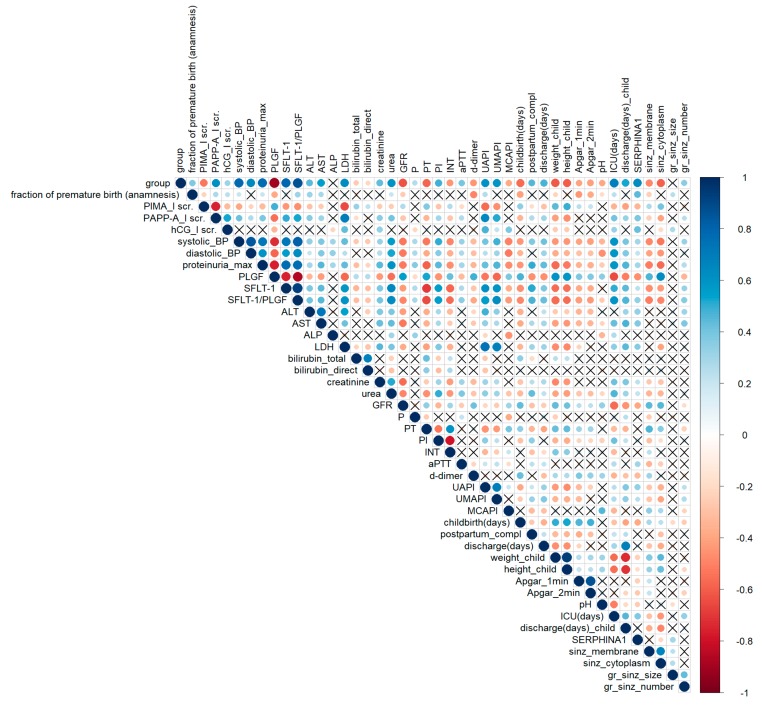
Spearman’s correlation analysis of clinical data, the expression level of *SERPINA1* in the urine and placenta at a confidence level of 0.05. A positive correlation is highlighted in blue, and a negative correlation is highlighted in red. The degree of correlation is highlighted in color—the stronger the correlation, the darker the color. “X”—statistically insignificant data. The parameter groups correspond to an increase in the severity of the pathology: 0—control, 1—chronic arterial hypertension (CAH), 2—PE-CAH, 3—mild PE, 4—severe PE. Premature—fraction of premature birth (anamnesis), I scr—first screening, BP—blood pressure, hCG—human chorionic gonadotropin, GFR—glomerular filtration rate, UAPI—uterine artery pulsatility index, UMAPI—fetal umbilical artery pulsatility index, MCAPI—fetal middle cerebral artery pulsatility index, ICU—intensive care unit (days), discharge—postpartum hospitalization (days), discharge_c—child postpartum hospitalization (days), childbirth—delivery time (days), complications—postpartum complications, PT—prothrombin time, PI—prothrombin index (hemostasiogram), INT—international normalized ratio (hemostasiogram), aPTT—activated partial thromboplastin time, uSER—average normalized intensity of *SERPINA1* peptides in urine, SER_mem/SER_cyto—intensity of *SERPINA1* staining of syncytiotrophoblast (SCT) membrane/cytoplasm, SER_size/SER_num—size and number of *SERPINA1* granules in SCT cytoplasm.

**Figure 3 ijms-21-00914-f003:**
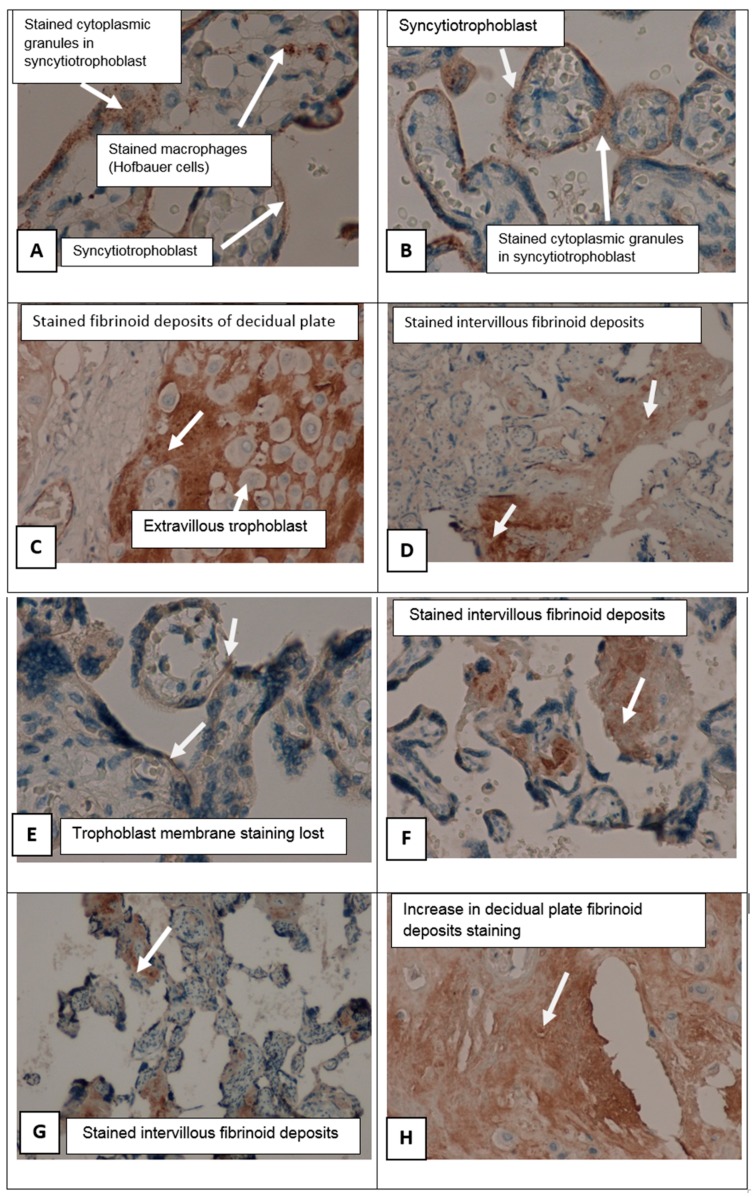
*SERPINA1* expression in placenta samples in cases of preeclampsia. (**A**)–(**D)** Placenta villi in PE in the group with no *SERPINA1* peptides in urine, stained granules, and membrane expression were determined. Fibrinoid deposits were stained, as well. (**A**,**B**) ×400; (**C**,**D**) ×200. (**E**)–(**H**) Placenta villi in PE in the group with *SERPINA1* peptides in urine; membrane expression was lost. Fibrinoid deposits were stained also. (**E**) ×400; (**F**) ×200; (**G**) ×100; (**H**) ×200.

**Figure 4 ijms-21-00914-f004:**
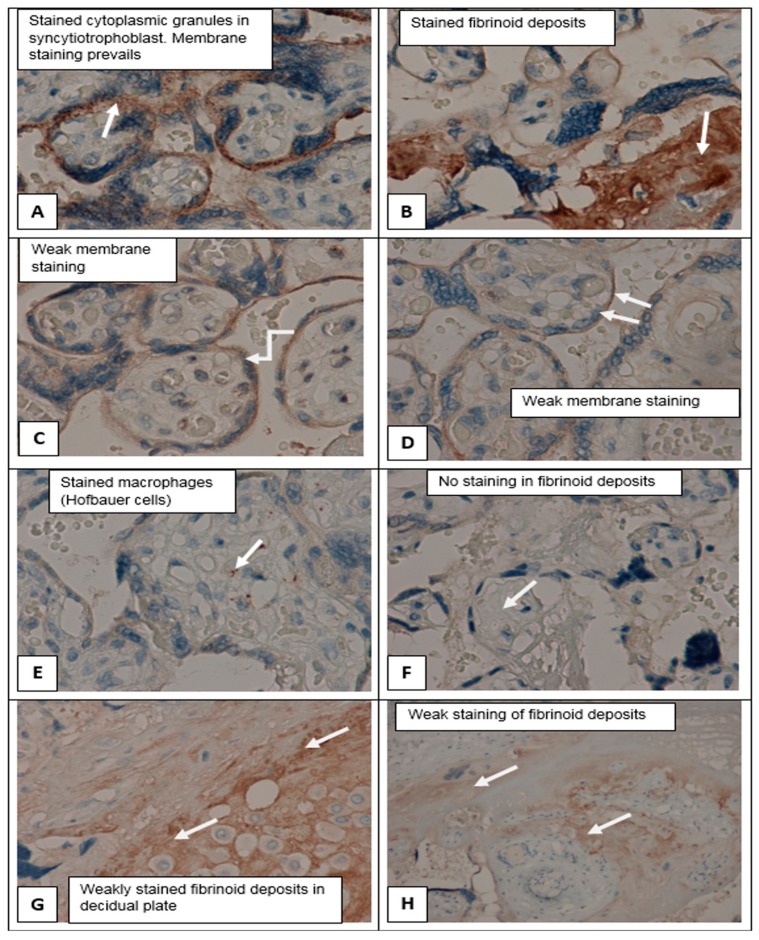
*SERPINA1* expression in placenta samples in cases of chronic arterial hypertension and uncomplicated pregnancy. (**A**,**B**) Placenta villi in CAH group; membrane staining predominated. Fibrinoid deposits were stained. (**A**,**B**) ×400. (**C**)–(**H**) Placenta villi in uncomplicated full-term pregnancy. Weak staining of syncytiotrophoblast, predominantly membrane staining; fibrinoid deposits were not stained or weakly stained; (**C**)–(**F**) ×400; (**G**) ×200; (**H**) ×100.

**Figure 5 ijms-21-00914-f005:**
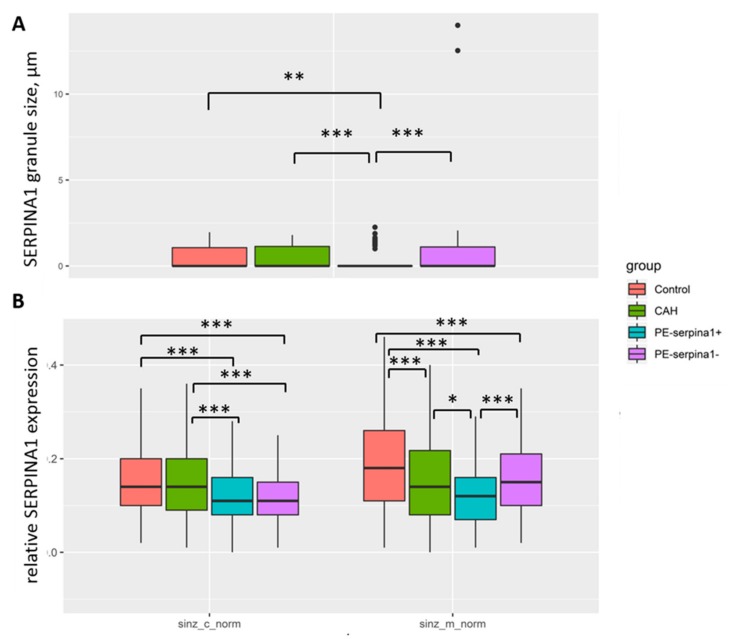
Comparison of *SERPINA1* expression in syncytiotrophoblast according to immunohistochemistry: (**A**) *SERPINA1* granule sizes (μm); (**B**) the relative level of expression of *SERPINA1* in the membrane and cytoplasm of syncytiotrophoblast. The results are given for the four study groups. *—*p* < 0.5; **—*p* < 0.01; ***—*p* < 0.001.

**Table 1 ijms-21-00914-t001:** Demographic and clinical data of the patients (*p*-values less than 0.05 are in bold).

Parameters	Mean (SD), *n* (%), Median (IQR)			*p*-Value
	Control (C), *n* = 18	CAH, *n* = 17	PE, *n* = 44	C vs. CAH C vs. PE CAH vs. PE
Age, years	31.8 ± 5	32 ± 4.6	30 ± 4.7	0.953 0.217 0.155
Body mass index	25 ± 2.8	29.8 ± 3.7	27.9 ± 5.5	**<0.001** **0.025** 0.135
Number of pregnancies	2.4 ± 1.2	2.2 ± 1.1	2.1 ± 1.4	0.620 0.100 0.356
Primiparous	10 (55.5)	8 (47.1)	31 (70.4)	0.620 **0.006** **0.055**
History of PE	1 (5.6)	4 (23.5)	8 (18.2)	**0.053** 0.076 0.715
Proteinuria, g/L	0.002 ± 0.001	0.039 ± 0.007	2.03 ± 1.79	**0.003** **<0.001** **<0.001**
Systolic blood pressure, mmHg	114.03 ± 7.67	141.16 ± 8.61	154.82 ± 9.37	**<0.001** **<0.001** **<0.001**
Diastolic blood pressure, mmHg	72.67 ± 6.59	93.55 ± 4.37	99.39 ± 8.29	**<0.001** **<0.001** **0.018**
Average blood pressure, mmHg	84.45 ± 6.87	109.42 ± 4.67	117.86 ± 8.21	**<0.001** **<0.001** **<0.001**
Cesarean section	10 (55.5)	10 (59.2)	33 (75)	0.806 0.195 0.132
Apgar 7 points or less, 1 min	2 (11.1)	4 (23.5)	19 (43.2)	0.314 **0.005** 0.121
Apgar 7 points or less, 5 min	0	1 (5.9)	9 (20.5)	- - 0.148
Newborns’ treatment in ICU	0	2 (11.8)	17 (38.6)	- - **0.033**

**Table 2 ijms-21-00914-t002:** Clinical features of patients of the PE_SER+ group in comparison with PE_SER− group (*p*-values less than 0.05 are in bold).

	Mean (SD), *n* (%), Median (IQR)		*p*-Value
	PE_SER+, *n* = 23	PE_SER1−, *n* = 21	
Systolic blood pressure, mmHg	154.76 ± 10.30	148.52 ± 9.55	**0.0099**
Diastolic blood pressure, mmHg	98.33 ± 8.17	96.52 ± 8.00	0.350
Average blood pressure, mmHg	117.14 ± 8.58	113.86 ± 7.86	0.096
Proteinuria, g/L	2.67 ± 1.20	1.44 ± 1.12	**0.0064**
Proteinuria, >3 g/L	9 (39.1)	2 (9.5)	0.010
Fetal growth restriction	4 (17.4)	5 (23.8)	0.610
Early PE	5 (21.7)	5 (23.8)	0.874
Severe PE	17 (74.0)	6 (28.5)	**0.0020**
sFLT/PLGF	130.93 (112.52)	192.52 (100.50)	0.474

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
