# Peer review of "SERPINA1 Peptides in Urine as A Potential Marker of Preeclampsia Severity"

_ijms, 2020, doi:10.3390/ijms21030914_

Round 1

Reviewer 1 Report

The manuscript focuses on one of the most important worldwide obstetric health problems, i.e. preeclampsia (PE). The study aim was to investigate the diagnostic potential of SERPINA1 peptides in the urine of pregnant patients with hypertensive disorders of pregnancy. This goal seems to be very interesting, however obtained results of the urinary peptides levels can be associated with kidney damage in the disease course.

Moreover, there are many limitations:

1) The authors should divide their study patients into five groups: control, chronic hypertension, preeclampsia superimposed on chronic hypertension, and two groups of preeclampsia (mild and severe) (or into six groups: as above, + the HELLP syndrome group). Considering the severity of PE is related to the topic of the manuscript: " SERPINA1 peptides in urine as a potential marker of preeclampsia severity".

The authors informed that: "PE diagnosis was established by the following symptoms: (...) liver impairment (increasing of alanine aminotransferase (ALT) and aspartate aminotransferase (AST) more than two times), (...); thrombocytopenia (<100x109 /l).." How many patients were diagnosed with HELLP syndrome?

2) Please specify the inclusion criteria of PE (the name of an international society).

3) Table 1: How can you explain that 59.2% of healthy pregnant patients (control group) delivered by cesarean sections.

4) Table 1: What means "3 (11.1)" in the control group in "Apgar 1 min". All results in "Apgar 1 min and 5 min" are difficult to understand.

5) Table 2: additional variables should be included:

gestational age during diagnosis of PE gestational age at taking of urine samples gestational age at the delivery birth weight of newborns treatment (including antihypertensives, magnesium sulphate, antenatal corticosteroids)

6) Figure 2 is not very readable. Please add an additional figure concerning the correlation between the expression level of Serphina 1 and sFLT/PLGF. 

7) There are some abbreviations in the manuscript, which can be explained (for example PSG2, 2-D LC-MS/MS).

8) Please use "Instruction for Authors" for all references.

Author Response

Reviewing: 1

The manuscript focuses on one of the most important worldwide obstetric health problems, i.e. preeclampsia (PE). The study aim was to investigate the diagnostic potential of SERPINA1 peptides in the urine of pregnant patients with hypertensive disorders of pregnancy. This goal seems to be very interesting, however obtained results of the urinary peptides levels can be associated with kidney damage in the disease course.

Answer: We are grateful that the reviewer noted this fact and completely agree with him. Moreover, the result obtained in the article does not contradict the general concept of PE development: the appearance of SERPINA1 peptides in the urine is detected in more severe cases of preeclampsia, which is naturally accompanied by more severe organ damage, and often multiple organ dysfunction. The associated significant change in the expression and localization of SERPINA1 in the placental trophoblast provides a new insight into the pathogenesis of preeclampsia and partially explains the origin of these peptides in the urine. We cannot associate the appearance of SERPINA1 peptides in urine only with impaired renal function, since an inverse correlation between the level of SERPINA1 peptides in urine and glomerular filtration rate was found. In addition, the size of the SERPINA1 peptides allows them to cross the renal filtration barrier. We do observe some SERPINA1 peptides in norma. But PE is characterized by certain urinary peptides SERPINA1 (Figure 1, Table 1S). This confirms the diagnostic significance of the SERPINA1 peptides in urine.

The authors should divide their study patients into five groups: control, chronic hypertension, preeclampsia superimposed on chronic hypertension, and two groups of preeclampsia (mild and severe) (or into six groups: as above, + the HELLP syndrome group). Considering the severity of PE is related to the topic of the manuscript: "SERPINA1 peptides in urine as a potential marker of preeclampsia severity".

Answer: Thank you for a valuable remark. We added the information about PE severity (severe/mild), form (early/late) : “Severe PE was diagnosed for 23 women, 10 women had early form of disease.” Due to the small number of patients in subgroups (PE with CAH - 8 patients) and a significant deterioration in the visual perception of information with a large (5 groups) number of groups (especially when presenting p - value), it was decided to refrain from additional separation in Tables 1 and 2. However, following the reviewer's advice, changes were made to the table where these subgroups were taken into account when analyzing the data.

Moreover, to prove the value of SERPINA1 urinary peptides in PE severity diagnosis we changed the group parameter in correlation analysis (Figure 2) to access the degree of disease: 0 - control, 1 - CAH, 2 - PE-CAH, 3 – mild PE, 4 – severe PE. The level of urinary SERPINA1 peptides positively correlated with the disease severity (parameter - group) with rS=0.48. It confirms our hypothesis and the article title “SERPINA1 peptides in urine as a potential marker of preeclampsia severity”. The text and Figure 2 was changes accordingly: “The group parameter corresponds to the severity of the woman's condition: 0 - control, 1 - CAH, 2 - PE-CAH, 3 – mild PE, 4 – severe PE.  This parameter correlates with almost all significant clinical and laboratory indicators of hypertensive pregnancy disorders (Fig. 4). Moreover, the level of urinary SERPINA1 peptides positively correlated with the disease severity (parameter - group) with rS=0.48. The opposite tendency was found for SERPINA1 expression in placenta (rS=-0.56). This confirms our hypothesis concerning the diagnostic potential of urinary SERPINA1 peptides.”

The authors informed that: "PE diagnosis was established by the following symptoms: (...) liver impairment (increasing of alanine aminotransferase (ALT) and aspartate aminotransferase (AST) more than two times), (...); thrombocytopenia (<100x109 /l).." How many patients were diagnosed with HELLP syndrome?

Answer: In the Materials and methods we included all criteria for preeclampsia in accordance with ACOG 2002. However, there were no patients with HELLP syndrome in our study.

Please specify the inclusion criteria of PE (the name of an international society). 

Answer: We used criteria for diagnosis preeclampsia of ACOG. (ACOG practice bulletin. Diagnosis and management of preeclampsia and eclampsia. Number 33, January 2002. Obstet Gynecol. 2002 Jan;99(1):159-67). The appropriate text was added.

Table 1: How can you explain that 59.2% of healthy pregnant patients (control group) delivered by cesarean sections.

Answer: There is indeed a high rate of caesarean section in our study group.  We supposed that it would be correct to compare the groups with similar methods of delivery (in particular, percent of caesarean section), so we collected women for control group with indication for caesarean, but without serious morbidity (such as: breech presentation, previous caesarean section scar, narrow pelvis). Moreover, all placental samples must be taken under similar conditions. It is necessary that there is the same level of hypoxia, etc. We included patients with high myopia in the control group, because this disease does not affect development of the placenta, uterine blood supply.

Table 1: What means "3 (11.1)" in the control group in "Apgar 1 min". All results in "Apgar 1 min and 5 min" are difficult to understand.

Answer: Thank you for this question. Digits in line Apgar (3(11.1) and other) mean absolute number and percentage in brackets. We made a mistake not indicating that this is a number of cases with Apgar score 7 points and less. Necessary adjustment was carried out in the table. Sorry for this mistake.

Table 2: additional variables should be included: gestational age during diagnosis of PE gestational age at taking of urine samples gestational age at the delivery birth weight of newborns treatment (including antihypertensives, magnesium sulphate, antenatal corticosteroids)

Answer: Thank you for appropriate suggestion. Additional variables were included to the table 2: gestational age at the moment of diagnosis of PE and at taking of urine samples were the same (urine samples were received at the moment of admission), gestational age at the delivery, birth weight. All patients received appropriate treatment with antihypertensive drugs, magnesium sulfate, glucocorticoids according to indications. However, the treatment was carried out after collecting urine samples (information included in the Material and Methods section).

Figure 2 is not very readable. Please add an additional figure concerning the correlation between the expression level of Serphina 1 and sFLT/PLGF.

Answer: Thank you. Figure 2 was edited to increase the letters size and parameters name were shorted. We hope it will make figure more readable The correlation between the expression level of Serphina 1 and sFLT/PLGF is one point on the figure 2.

There are some abbreviations in the manuscript, which can be explained (for example PSG2, 2-D LC-MS/MS). 

Answer: We explained the abbreviations in the manuscript.

Please use "Instruction for Authors" for all references.

Answer: References were corrected.

Reviewer 2 Report

The manuscript "SERPINA1 peptides in urine as a potential marker of preeclampsia severity" described an interesting, well-designed and well-performed study on the validity of screening such peptides as a tool for making decisions on preeclampsia cases. There is no major concerns on that. However, the text is flawed by a too verbose writing style and I think that a deep revision of the English writing style would notably benefit the paper. The same applies for the abstract, where it is difficult, like in the main text, to discern between SERPINA and SHERPINA, as an example. 

Author Response

Reviewing: 2

The manuscript "SERPINA1 peptides in urine as a potential marker of preeclampsia severity" described an interesting, well-designed and well-performed study on the validity of screening such peptides as a tool for making decisions on preeclampsia cases. There is no major concerns on that. However, the text is flawed by a too verbose writing style and I think that a deep revision of the English writing style would notably benefit the paper. The same applies for the abstract, where it is difficult, like in the main text, to discern between SERPINA and … 

Answer:

Thank you for the positive review. We have corrected the entire text as much as possible.

Round 2

Reviewer 1 Report

Dear Authors,

Thank you for attempting to address my concerns.

I accept the manuscript in the present form.

Author Response

Thank you very much!

Reviewer 2 Report

I cannot see the changes in the manuscript since they are not highlighted or marked using changes track but I recommend revision of the English grammar in the manuscript by professional services.

Author Response

Thank you very much for your comments.  English grammar in the manuscript  was eddited by professional (MDPI) services.

Round 3

Reviewer 2 Report

The manuscript can now be accepted for publication, from my point of view

Author Response

Thank you very much for your comments.